# Transcranial Direct Current Stimulation (tDCS) for Borderline Personality Disorder (BPD): Why and How?

**DOI:** 10.3390/brainsci15060547

**Published:** 2025-05-23

**Authors:** Lionel Cailhol, Kamilia Soltani, Cécilia Neige, Marine Mondino, Jérôme Brunelin, Martin Blay

**Affiliations:** 1Faculté de Médecine, Département de Psychiatrie et D’addictologie, Pavillon Roger-Gaudry, C.P. 6128, Succursale Centre-Ville, Montréal, QC H3T 1J4, Canada; kamilia.soltani@umontreal.ca; 2Centre de Recherche de L’Institut Universitaire de Santé Mentale de Montréal, 7401 Rue Hochelaga, Montréal, QC H1N 3M5, Canada; 3CERVO Brain Research Center, 2301 Av. D’Estimauville, Québec City, QC G1E 1T2, Canada; 4CIUSSS de L’Est de L’île de Montréal, Montréal, QC H1A 1T5, Canada; 5PsyR2 Team, U1028 UMR5292, Centre de Recherche en Neurosciences de Lyon CRNL, CNRS, INSERM, Université Claude Bernard Lyon 1, 69500 Bron, France; cecilia.neige@ch-le-vinatier.fr (C.N.); marine.mondino@ch-le-vinatier.fr (M.M.); jerome.brunelin@ch-le-vinatier.fr (J.B.); 6PsyR2, Le Vinatier Psychiatrie Universitaire Lyon Métropole, 95 Boulevard Pinel, 69500 Bron, France; 7ADDIPSY, Santé Basque Développement Group, Addictology and Psychiatry Oupatient Center, 69007 Lyon, France; martin.blay5@gmail.com; 8Centre de Recherche en Epidémiologie et Santé des Populations Team “DevPsy”, INSERM, UVSQ, Université Paris-Saclay, 94807 Villejuif, France

**Keywords:** borderline personality disorder, tDCS, emotional dysregulation, impulsivity

## Abstract

Background: Borderline Personality Disorder (BPD) is a severe psychiatric condition characterized by pervasive emotional dysregulation, impulsivity, and unstable interpersonal relationships. Affecting over 1% of the general population, BPD carries significant morbidity, frequent hospitalizations, and an increased risk of suicide. Although specialized psychotherapeutic approaches have shown efficacy, their impact is often constrained by availability, lengthy treatment durations, moderate effect sizes, and high dropout rates. Pharmacological treatments for BPD remain inadequate and are usually accompanied by adverse side effects. Objective: This narrative review seeks to explore the potential of transcranial direct current stimulation (tDCS) as a safe, cost-effective, and accessible neuromodulation intervention aimed at alleviating core BPD symptoms—namely, emotional dysregulation and impulsivity—while also addressing common comorbidities and opportunities for integration with existing therapeutic modalities. Methods: We conducted a narrative literature synthesis in accordance with the SANRA (Scale for the Assessment of Narrative Review Articles) guidelines. A PubMed/MEDLINE search was performed using keywords related to transcranial direct current stimulation (tDCS) and BPD, identifying five published randomized controlled trials on the topic. To provide a broader perspective, we also included studies from related fields examining mechanisms of action, safety and tolerability, cost-effectiveness, stimulation parameters, and clinical outcomes relevant to BPD. Results: Conventional tDCS protocols—typically involving 1–2 mA currents for 20–30 min—have demonstrated an excellent safety profile, resulting in only minimal and transient side effects without any risk of overdose or misuse, which is a key advantage for populations at high risk of suicidality. With moderately priced devices and the feasibility of home-based administration, tDCS provides a substantially more affordable alternative to both long-term pharmacotherapy and intensive psychotherapy. Neurobiologically, tDCS modulates the excitability of the dorsolateral and ventrolateral prefrontal cortex and enhances fronto-limbic connectivity, thereby strengthening top-down regulatory control over emotion and behavior. Pilot randomized controlled trials report moderate effect sizes for improvements in emotional regulation, inhibitory control, and rejection sensitivity, along with ancillary gains in executive functioning and reductions in depressive and substance-use symptoms when stimulating the left dorsolateral prefrontal cortex. Conclusions: tDCS stimulation emerges as a safe and scalable adjunctive treatment for BPD, leveraging targeted neuromodulation to address core features and common comorbidities like depression. However, variability in current protocols and the scarcity of well-powered randomized trials underscore the pressing need for standardized methodologies, longer-term follow-up, and individualized stimulation strategies to establish enduring clinical benefits.

## 1. Introduction

Borderline Personality Disorder (BPD) is a severe and complex psychiatric condition characterized by pervasive emotional dysregulation, impulsivity, and interpersonal dysfunction [1]. It affects more than 1% of the general population [2] and is associated with high rates of psychiatric comorbidities, particularly mood disorders, anxiety disorders, and substance use disorders [3]. Beyond its clinical features, BPD is among the most impairing psychiatric disorders, leading to significant functional impairment [4], frequent hospitalizations [5], and an elevated risk of suicide [6,7]. Individuals with BPD experience a mortality rate substantially higher than that of the general population, primarily due to suicide and related health complications [6,8]. Given its prevalence and severity, developing effective and accessible treatment strategies remains a critical challenge in psychiatry.

Specialized psychotherapy is currently the gold standard for BPD treatment [9,10,11]. Approaches such as Dialectical Behavior Therapy (DBT), Mentalization-Based Therapy (MBT), and Schema Therapy have demonstrated efficacy in reducing emotional instability, impulsive behaviors, and self-harm [12]. Despite their effectiveness, these treatments face considerable limitations. The demand for specialized therapy far exceeds the availability of trained clinicians, resulting in limited access to care and prolonged wait times [13]. Even when accessible, psychotherapy requires long-term engagement; while many patients benefit from these interventions, the effect sizes remain moderate [12], with improvements often occurring gradually over months or years. A further challenge is that individuals with BPD frequently struggle with treatment adherence [14], resulting in high dropout rates and diminished therapeutic benefits. In parallel, pharmacotherapy for BPD remains unsatisfactory. Although antidepressants, mood stabilizers, and antipsychotics are widely prescribed, no pharmacological treatment has been explicitly approved for BPD [15]. Available medications offer only partial symptom relief and frequently come with significant side effects, complicating treatment decisions. These challenges underscore the urgent need for innovative, alternative, or adjunctive approaches to improve treatment strategies and patient outcomes.

Neuromodulation has emerged as a promising avenue for psychiatric disorders, especially BPD, offering novel mechanisms to modulate brain activity and alleviate symptoms [16,17]. Transcranial direct current stimulation (tDCS) has gained attention among available techniques due to its safety, accessibility, and potential therapeutic effects. Notably, it is more cost-effective than repetitive transcranial magnetic stimulation (rTMS), and its potential for home-based administration makes it an attractive option for patients who face barriers to accessing traditional in-clinic treatments [18]. In the context of BPD, tDCS may be particularly relevant given its potential capacity to modulate neural networks implicated in emotion regulation and impulsivity. One of the most common hypotheses is the fronto-limbic hypothesis [19], positing that emotional dysregulation arises from heightened amygdala activity, reduced activation in prefrontal regions such as the dorsal anterior cingulate cortex, and weakened fronto-limbic connectivity [20,21], although the validity of this hypothesis remains debated by some authors [22]. By targeting these neural mechanisms, tDCS has the potential to address core symptoms of BPD in a biologically plausible and clinically meaningful manner.

In this context, this narrative review aims to explore the potential role of tDCS in the treatment of BPD. Specifically, it will examine the rationale for integrating tDCS into BPD care—focusing on its safety, affordability, and capacity to target key symptom domains such as emotional dysregulation and impulsivity—and will discuss optimal stimulation protocols (electrode placement, intensity, session duration, and frequency) as well as its potential to complement or enhance psychotherapy. We ask whether tDCS can safely, affordably, and effectively modulate core BPD features, and we hypothesize that (1) specific tDCS parameters (e.g., electrode montage and stimulation intensity) will differentially influence symptomatic targets, from core BPD symptoms to executive functions and common comorbidities, and (2) tDCS will provide additional benefits in terms of cost-effectiveness and synergistic gains when combined with psychotherapeutic interventions. Finally, we will address clinical applications and feasibility, evaluate tDCS as both a standalone and adjunctive treatment, highlight current gaps in the literature, and outline future research directions.

## 2. Materials and Methods

A comprehensive narrative review was conducted to synthesize preclinical and clinical evidence on tDCS for BPD. This narrative review was conducted in accordance with the guidelines of the SANRA (Scale for the Assessment of Narrative Review Articles), which outlines quality criteria for narrative reviews. The manuscript adheres to SANRA’s six domains: justification of the article’s importance, clear statement of objectives, description of the literature search, appropriate referencing, sound scientific reasoning, and proper presentation of data [23]. We searched PubMed/MEDLINE from database inception through 30 March 2025, using combinations of the keywords “borderline personality disorder”, “BPD”, “transcranial direct current stimulation”, “tDCS”, “neuromodulation”, “prefrontal”, “emotion regulation”, and “impulsivity.” No language restrictions were applied. Reference lists of included articles and pertinent review papers were hand-searched for additional studies. The review process was carried out in two steps. First, we focused on identifying RCTs investigating tDCS in individuals with BPD. Data extraction was conducted independently by two reviewers (MB, LC), based on a prior unpublished systematic review conducted by our team. Second, we performed targeted searches to complement this initial synthesis by including studies addressing (1) the efficacy of tDCS in treating common BPD comorbidities (identified through recent meta-analyses), (2) the safety profile of tDCS (via meta-analyses and clinical guidelines), and (3) cost-related aspects (medico-economic evaluations). Finally, we incorporated selected studies to enrich our understanding of the neurobiological mechanisms underlying tDCS and its potential synergies with psychotherapeutic interventions.

## 3. Results

### 3.1. tDCS

#### 3.1.1. Description

tDCS is a non-invasive neuromodulation technique that delivers a low-intensity electrical current (typically 1–2 mA) through electrodes placed on the scalp, usually for sessions lasting around 20 min [24]. Depending on the polarity, tDCS can modulate cortical excitability: anodal stimulation generally increases neuronal activity, while cathodal stimulation tends to decrease it. Although tDCS does not induce action potentials directly, it alters the resting membrane potential, thereby influencing the likelihood of neuronal firing and facilitating synaptic plasticity. These effects are thought to underlie its potential to modulate cognitive, emotional, and behavioral processes in psychiatric conditions such as borderline personality disorder.

#### 3.1.2. Mechanisms of Action

Beyond the local neurophysiological effects in the cortical regions beneath the stimulation electrodes, tDCS effects also spread to interconnected cortical areas, modulating the connectivity of resting-state networks [25,26]. This effect can extend to deeper brain regions, specifically stimulating cortico-subcortical loops and modulating mesocorticolimbic dopaminergic transmission when stimulation is applied to the prefrontal cortex [27]. The neurobiological effects of tDCS may also extend beyond its impact on the brain; for instance, it helps regulate stress reactivity by reducing cortisol secretion and preventing cognitive changes induced by acute stress [28].

### 3.2. Rationale for Using tDCS

#### 3.2.1. Safety Profile of tDCS in Comparison to Pharmacotherapy

One of the most compelling reasons to explore tDCS as a treatment for BPD is its favorable safety profile compared to psychopharmacological interventions [29,30]. A 2016 review of the current literature, encompassing more than 1000 participants and totaling over 33,000 sessions, concluded that conventional tDCS protocols used in human trials (≤40 min, ≤4 milliamps, ≤7.2 Coulombs) have not been associated with any serious adverse effects or irreversible injuries [30]. Medications frequently prescribed to individuals with BPD [31], including antidepressants, mood stabilizers, and antipsychotics, are linked to significant side effects (such as weight gain, sedation, metabolic disturbances, and cognitive impairment), despite having limited efficacy in addressing core BPD symptoms, which often leads to polypharmacy without clear benefits [15,31].

Beyond tolerability, safety concerns related to self-harm and suicide risk are particularly relevant in BPD [10]. A significant proportion of individuals with BPD engage in suicidal behaviors and self-injury, with an estimated 6 to 8% ultimately dying by suicide [7,32]. Commonly prescribed medications, such as benzodiazepines, may increase behavioral dyscontrol and worsen impulsivity [33], while other medications, like lithium or tricyclic antidepressants, present a heightened risk in this population due to their potential for a lethal overdose [34]. In contrast, tDCS does not pose a risk for overdose or misuse, making it a potentially safer alternative or adjunctive treatment for individuals with high suicidality. The mild side effects of tDCS, including transient itching, tingling, or mild headaches, are generally well tolerated and do not significantly impact daily functioning. This favorable tolerability and safety profile makes tDCS a viable option, particularly for patients who do not respond well to medication or who are at risk of self-harm with pharmacological interventions.

#### 3.2.2. Affordability and Accessibility Compared to Conventional Treatments

Cost-effectiveness may be another significant advantage of tDCS compared to traditional treatment approaches. In Denmark, a study analyzing societal costs estimated that the total direct healthcare expenses and productivity losses for individuals with BPD amounted to EUR 40,441 [35]. Although widely utilized, pharmacotherapy also imposes a substantial financial burden on healthcare systems and patients. The chronic use of multiple medications, which is common in BPD management, results in high cumulative costs, particularly in cases requiring frequent dose adjustments or management of adverse effects. The average annual cost of medication for BPD patients is estimated at EUR 1589 [35].

In contrast, tDCS represents a relatively low-cost intervention. The price of a single tDCS device, starting at EUR 300, is significantly lower than the total expenses associated with long-term medication use or intensive psychotherapy [36,37]. Additionally, tDCS allows for home-based administration [38], which can help reduce costs linked to frequent clinical visits and enhance accessibility, especially for individuals in remote or underserved areas. For a full course of 15 hospital-based tDCS sessions, Sauvaget estimated the cost to be EUR 1555 per patient [36]. In comparison, Le Bars et al. reported that 10 sessions of home-based tDCS cost approximately EUR 231.74 [39]. This highlights the substantial economic advantage of home-based delivery.

It is important to note that existing economic evaluations of tDCS have been conducted in only a few countries—primarily France and Canada—and, therefore, cannot be readily generalized to an international context. There is a clear lack of global data. Moreover, tDCS devices typically require approval from national regulatory authorities, which can lead to substantial variability in the availability, cost, and implementation across countries. The current literature does not address these discrepancies. Device pricing may differ significantly depending on the region, and the overall cost-effectiveness of tDCS is closely tied to the structure and resources of the local healthcare system. As such, caution is warranted when interpreting or extrapolating cost-effectiveness data for broader use.

#### 3.2.3. Neurobiological Basis for tDCS in BPD

The application of tDCS in BPD is backed by a growing body of evidence highlighting neurobiological dysfunctions in this disorder, particularly in the circuits involved in emotion regulation, impulse control, and executive function. The DLPFC has been recognized as a key region implicated in the pathophysiology of BPD. Neuroimaging studies have consistently shown reduced activity and structural abnormalities in the DLPFC, especially regarding its role in modulating limbic hyperactivity. Dysfunctional connectivity between the DLPFC and regions such as the amygdala and anterior cingulate cortex contributes to the emotional instability, impulsive decision making, and interpersonal difficulties that characterize BPD.

tDCS offers a direct method of modulating DLPFC activity, enhancing prefrontal regulation over subcortical structures involved in emotional and behavioral control, including DA release and stress reactivity. The anodal stimulation of the left DLPFC has been associated with improved emotional regulation and cognitive control, while the anodal stimulation of the right DLPFC may further reduce impulsivity and aggression. By leveraging these neuromodulatory effects, tDCS represents a targeted, neurobiologically informed intervention for BPD that aligns with existing models of prefrontal–limbic dysfunction in the disorder.

### 3.3. Possible Outcomes

#### 3.3.1. Targeting BPD Symptoms

Emotional dysregulation is a core symptom of BPD, contributing to the intense mood swings, anger outbursts, and interpersonal instability that define the disorder [40]. As mentioned earlier, the fronto-limbic hypothesis states that emotional dysregulation could be linked to deficiencies in top-down control, i.e., the prefrontal hyporegulation of limbic hyperactivation. This hypothesis recalls the mechanisms of action of current psychotherapeutic interventions, such as DBT, which focus heavily on improving emotion regulation through learning distress tolerance, communication, and cognitive restructuring (thus, through increasing top-down control), even though not all patients respond adequately to these approaches [41]. Given the potential role of prefrontal hypoactivity in impaired emotional control [19], tDCS may offer a means to enhance the cognitive regulation of emotions by increasing the prefrontal excitability. Most studies on tDCS in BPD have investigated its effect on hypothesized cognitive control over emotions (Table 1). Preliminary empirical evidence suggests that prefrontal stimulation is effective in alleviating emotional dysregulation among patients with BPD [42,43]. As an illustration, we calculated the effect size for emotion regulation (measured by the Emotion Regulation Questionnaire) in the Molavi study, which yielded a very large effect (Cohen’s d = 4). However, it remains unclear whether this improvement translates to clinical improvements in emotion regulation scores [16,44].

Impulsivity, another hallmark feature of BPD, is linked to deficient inhibitory control mechanisms within the prefrontal cortex [19]. Poor impulse regulation contributes to self-harm, reckless behaviors, and difficulties in maintaining stable relationships or employment [10]. Evidence from studies on tDCS among healthy individuals and various psychiatric disorders suggest that enhancing the prefrontal function may improve impulse control [45,46]. Similar effects were observed in BPD using DLPF stimulation, which improved the decision-making process with a moderate effect size [44]. As an illustration, we calculated the effect size for impulsivity, measured by the Barratt Impulsiveness Scale (BIS-11), in the Lisoni study, which showed a large effect (Cohen’s d = 1.12).

Finally, a recent study has opened the possibility that tDCS may influence the interpersonal dimension of BPD, specifically, rejection sensitivity [47]. In this study, the anodal stimulation of the right ventrolateral prefrontal cortex (rVLPFC), compared to sham tDCS, resulted in a normalization of the rejection response as measured by the Cyberball task. As an illustration, we calculated the effect sizes for rejection-related emotions, measured by the Rejected Emotion Scale (RES), in the Lisco study. The results showed a large effect for inclusion (Cohen’s d = 0.95), a moderate effect for exclusion (Cohen’s d = 0.75), and no significant effect for overinclusion. This represents the first biological intervention demonstrated in an RCT to affect the relational dimension of BPD. This work aligns with current knowledge on the prosocial effect of tDCS [48].

Several studies failed to demonstrate an effect on their primary outcomes. This may be explained by various factors, including methodological limitations (e.g., underpowered designs), population heterogeneity (e.g., medication use, comorbidities, and severity levels), suboptimal stimulation parameters (e.g., dosage, electrode placement, and number of sessions), or the insufficient control of brain activity during stimulation.

**Table 1 brainsci-15-00547-t001:** RCTs assessing tDCS in BPD population.

Author	Participants	Anode	Cathode	Anodal Electrode Size (cm^2^)	Current Intensity (mA)	Number of Session	Session Duration (mn)	Main Effects
Schulze et al., 2019 [49]	48 BPD: 25 sham; 23 active	Right DLPFC	Left deltoid	35	1	1	20	No improvement of cognitive control over negative stimuli
Molavi et al., 2020 [42]	32 BPD: 16 sham; 16 active	Right DLPFC	Left DLPFC	25	2	10	20	Improvement of executive functions and cognitive emotion regulation, although emotional expression was unaffected
Lisoni et al., 2020 [44]	30 BPD: 15 sham; 15 active	Left DLPFC	Right DLPFC	35	2	15	20	Improvement of impulsivity, aggression, and craving, while also marginally improving decision making
Wolkenstein et al., 2021 [43]	40: 20 healthy; 20 BPD	Left DLPFC	Right Mastoïd	35	1	1	20	No reduction in emotional interference at the group level, but selectively improved cognitive control in participants—particularly BPD patients.
Lisco et al., 2025 [47]	40 BPD: 20 sham; 20 active	Right VLPFC	Left supraorbital region	25	1.5		20	Reduction in rejection-related emotions

#### 3.3.2. Enhancing Psychosocial Functioning Through Executive Function

Beyond its impact on some BPD symptoms, tDCS may also enhance executive functioning, which is a critical factor in psychosocial adaptation for individuals with BPD. Deficits in working memory, cognitive flexibility, and attentional control are frequently reported in BPD [50], contributing to challenges in employment, academic performance, and interpersonal relationships [4]. These cognitive impairments are primarily driven by dysfunction in the DLPFC and its broader executive control network [19], making it a potentially important target for neuromodulatory interventions.

tDCS over the DLPFC has been shown to enhance the executive function performance among BPD patients [42]. This aligns with evidence collected from the DLPFC stimulation of healthy people [51]. More specifically, the small size of the anode and the extracephalic placement of the cathode appear to amplify this effect. By addressing the underlying cognitive deficits contributing to psychosocial dysfunction, tDCS has the potential to enhance traditional psychotherapeutic interventions, providing a complementary approach to improving real-life outcomes for patients with BPD.

#### 3.3.3. Efficacy of tDCS in Comorbid Conditions

BPD is highly comorbid with major depressive disorder (MDD), anxiety disorders, substance use disorders, and medical conditions, which significantly worsen the overall prognosis [3,52], especially the suicide risk [32]. tDCS has been extensively studied in the treatment of depression, with strong evidence demonstrating its efficacy in alleviating depressive symptoms. Meta-analyses indicate that the anodal stimulation of the left DLPFC produces mild-to-moderate antidepressant effects [53,54,55]. Given that over 80% of individuals with BPD experience major depressive episodes [3], integrating tDCS into BPD care may not only address the core symptoms of the personality disorder, but also improve comorbid depression, thereby enhancing the overall treatment response.

Additionally, tDCS has shown promising effects in bipolar depression [56], substance use disorder [54], pain in fibromyalgia [57], hallucinations [58,59], Attention Deficit/Hyperactivity Disorder [60], and anxiety, including Post-Traumatic Stress Disorder [61]. It is worth noting that different stimulation parameters were applied across these various conditions. Nevertheless, BPD comorbidities remain important targets for intervention in individuals with borderline personality disorder.

### 3.4. What Is the Optimal tDCS Protocol for BPD?

#### 3.4.1. Stimulation Parameters

There is no single optimal stimulation protocol for patients with BPD, and multiple factors should be considered. Regarding core BPD symptoms, three anodal stimulation sites have been tested [42,44,47] (Figure 1), although they have not been directly compared. The anodal stimulation of the left DLPFC appears to impact emotional regulation and executive functioning [42]. In contrast, right DLPFC stimulation has shown effects on impulsivity [44] and rVLPFC stimulation has shown effects on rejection sensitivity [47]. While these findings still need to be confirmed, lateralization could be tailored to the patient’s predominant symptom profile. Concerning stimulation parameters, protocols typically involve 20 min sessions, with intensities ranging from 1 to 2 mA and 1 to 10 sessions.

When considering comorbidities, the anodal stimulation of the left DLPFC is most effective for depressive symptoms [53], while the right DLPFC may be more suitable for addressing craving [62]. Fibromyalgia-related symptoms, on the other hand, seem to respond to stimulation over the primary motor cortex (M1) [57]. Thus, the choice of stimulation site and parameters should be adapted to the patient’s comorbidity profile. Regarding stimulation parameters, protocols typically involve 20 to 30 min sessions, with intensities ranging from 1 to 2 mA. It is worth noting that current parameters for depression tend to use longer sessions and reinforcement sessions after the 2 to 3 weeks of initial treatments [18]. This also opens the possibility for home-based treatment.

#### 3.4.2. Combination with Psychotherapy

From a theoretical standpoint, combining tDCS with psychotherapy appears both feasible and potentially beneficial, particularly for enhancing early therapeutic effects, creating synergistic outcomes, and fostering hope in patients. However, caution is warranted in this area. To date, no published studies have investigated the combined use of tDCS and psychotherapy in BPD. Moreover, findings from other clinical domains have been relatively discouraging. For instance, in the treatment of depression, the combination of cognitive behavioral therapy and tDCS has not demonstrated superior efficacy compared to either treatment alone [63]. Similarly, pairing rTMS with DBT has not shown additional clinical benefit in BPD [64]. Therefore, it is essential to distinguish between the research setting, where identifying optimal synergies remains an open question, and clinical practice, where the combination of tDCS and psychotherapy cannot yet be regarded as a superior option.

### 3.5. Future Research Directions

#### 3.5.1. Caution Is Needed

According to Lisoni, the current evidence on Non-Invasive Brain Stimulation (NIBS) in BPD, including tDCS, is limited by the heterogeneity of the stimulation protocols, the lack of RCTs, and the generally poor quality of existing studies, which often present a high risk of methodological bias. These limitations significantly constrain the development of standardized, evidence-based clinical recommendations. To guide future systematic investigations, Lisoni proposes protocol optimization strategies, emphasizing the exploration of alternative stimulation targets and encouraging a symptom-based NIBS approach [16,17].

#### 3.5.2. Clinical Trials Needed

To date, only a limited number of studies have directly examined the effects of tDCS in individuals with BPD. Most available evidence stems from research on comorbid conditions such as depression, anxiety, or substance use disorders. There is a pressing need for RCTs specifically targeting BPD and its comorbidities, using rigorous methodologies, appropriate control conditions, and sufficiently powered samples. These trials should also aim to explore both symptom-specific and global functional outcomes.

#### 3.5.3. Long-Term Effects and Sustainability

A critical unanswered question is whether the clinical benefits of tDCS persist beyond the treatment period. Based on recent studies in patients with depression [38], we may hypothesize that longer-lasting effects are more likely to occur with an increased number of sessions. Longitudinal studies are needed to evaluate the durability of symptom improvement and the risk of relapse over time. Furthermore, research should explore whether repeated or maintenance sessions contribute to longer-lasting effects, possibly through mechanisms of neuroplasticity.

#### 3.5.4. Personalized Neuromodulation

Interindividual variability in response to tDCS suggests that a one-size-fits-all approach may be suboptimal. Future studies should explore the potential of personalized protocols informed by neuroimaging, electrophysiological data, or other biomarkers. Identifying patient subgroups based on their clinical phenotype, symptom dominance, or neurocognitive profile that are more likely to benefit from tDCS could significantly enhance the treatment efficacy and reduce unnecessary exposure to ineffective interventions.

#### 3.5.5. Psychotherapeutic Combination

Psychotherapy is a lengthy and complex process comprising multiple interacting components, characterized by considerable technical heterogeneity. Moreover, most psychotherapeutic treatments for BPD typically span several months or even years [11]. In contrast, neuromodulation interventions, such as tDCS, are usually delivered over a much shorter period, often lasting only a few weeks. To investigate potential synergies between tDCS and psychotherapy, we propose integrating a symptom-targeted approach. For example, activating emotionally regulated scenarios during the anodal stimulation of the left DLPFC may help enhance the neural circuitry underlying emotion regulation. In this context, practicing specific DBT emotion regulation skills, such as emotion labeling, during stimulation could strengthen this domain. Similarly, skills targeting impulsivity, such as the well-known DBT “STOP” technique, could be combined with the anodal stimulation of the right DLPFC to reduce impulsive behaviors. Moreover, considering the neuropsychological deficits associated with BPD, it may be beneficial to incorporate cognitive training modules focused on executive functions or working memory [65]. Finally, very brief psychotherapeutic interventions, such as guided imagery or imagery rescripting [40], could also be well-suited to the short duration of neuromodulation sessions and offer promising avenues for synergistic treatment.

## 4. Conclusions

tDCS represents a promising, safe, and cost-effective neuromodulation technique for addressing core symptoms of BPD, enhancing psychosocial functioning, and managing the most frequent comorbidities. Although the field is still in its infancy, the preliminary findings suggest that tDCS may offer meaningful benefits, particularly in modulating emotional dysregulation, impulsivity, and interpersonal sensitivity. We could state that while several comorbidities, such as depression, are supported by strong evidence that may justify the clinical implementation of tDCS, the effects on core BPD features such as impulsivity, affective instability, and rejection sensitivity remain preliminary and require confirmation through more robust, well-powered studies.

However, current evidence is limited by methodological heterogeneity, a lack of disorder-specific RCTs, and short-term follow-up. To advance the field, well-designed clinical trials with standardized protocols and rigorous outcome measures are urgently needed. These studies should also assess the durability of therapeutic effects and explore underlying neurobiological mechanisms.

Future research should explore the potential of combining tDCS with psychotherapeutic interventions and developing personalized stimulation protocols informed by clinical profiles and neurobiological markers. Such integrated approaches may enhance the treatment efficacy and help tailor interventions to meet the individual needs of patients with BPD.

## Figures and Tables

**Figure 1 brainsci-15-00547-f001:**
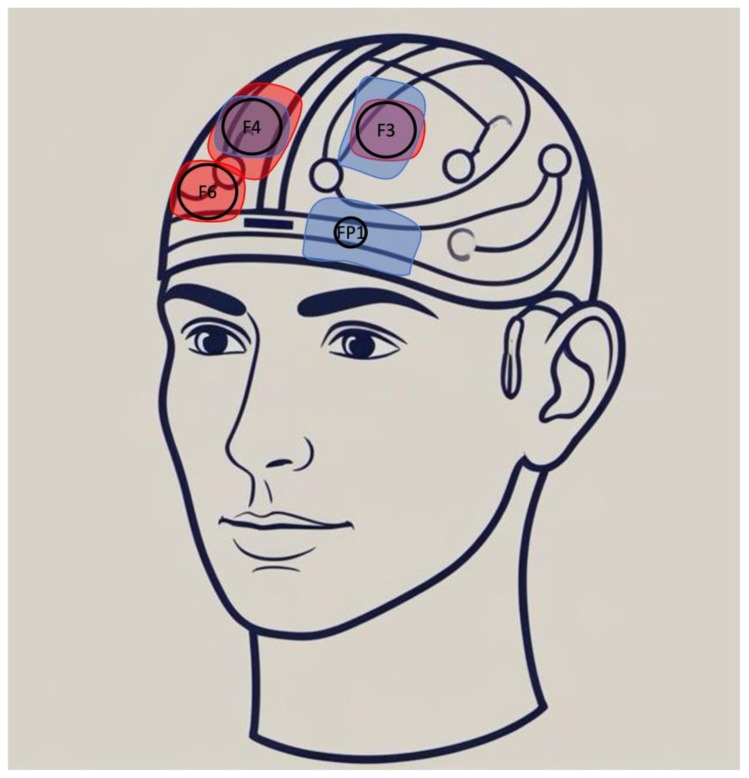
Examples of tDCS electrode montages in patients with BPD (anode in red, cathode in blue). Three montages are shown: anode F3/cathode F4 (25 cm^2^) targeting emotion dysregulation [42], anode F6 (25 cm^2^)/cathode FP1 (35 cm^2^) targeting rejection [47]. anode F4/cathode F3 (35 cm^2^) targeting impulsivity [44].

## Data Availability

This study did not create or analyze new data, so data sharing is not applicable.

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
