# Peer review of "Transcranial Direct Current Stimulation (tDCS) for Borderline Personality Disorder (BPD): Why and How?"

_brainsci, 2025, doi:10.3390/brainsci15060547_

Round 1
Reviewer 1 Report
Comments and Suggestions for Authors
This narrative review presents a timely and clinically significant exploration of the potential role of transcranial direct current stimulation (tDCS) in the treatment of Borderline Personality Disorder (BPD). The manuscript is grounded in a solid understanding of BPD’s neurobiological underpinnings and therapeutic challenges, offering a compelling rationale for considering tDCS as a low-cost, non-invasive neuromodulatory adjunct or alternative to traditional interventions.
The structure of the manuscript is coherent, with well-defined sections covering the pathophysiology of BPD, the neurophysiological mechanisms of tDCS, clinical safety, cost-effectiveness, and preliminary efficacy data. The authors provide a comprehensive and well-referenced synthesis of existing literature, making judicious use of both clinical and preclinical findings. The abstract is well-written and summarizes key points clearly. Moreover, the manuscript adheres to the SANRA guidelines for narrative reviews, enhancing its methodological transparency.
The discussion of tDCS mechanisms is neurobiologically sound and includes current evidence on the modulation of fronto-limbic circuits, stress reactivity, and executive function, all of which are relevant to BPD pathophysiology. The authors demonstrate a nuanced understanding of these systems and emphasize the differential impact of electrode montages (e.g., left vs. right DLPFC, rVLPFC), which is a strength of the paper. The figure illustrating common electrode placements is particularly helpful and enhances the clinical applicability of the review.
The emphasis on the safety profile of tDCS is highly appropriate, especially considering the elevated risk of self-harm and suicidality in the BPD population. The comparison to pharmacological alternatives is balanced and data-driven, acknowledging both the limitations of current pharmacotherapy and the absence of FDA-approved medications for BPD. Similarly, the cost analysis is clearly articulated and underscores the potential economic benefits of tDCS, particularly in home-based applications.
The manuscript’s analysis of tDCS efficacy for core BPD symptoms—emotional dysregulation, impulsivity, and rejection sensitivity—is both critical and well-supported. The authors appropriately qualify their conclusions given the scarcity of randomized controlled trials (RCTs) specifically targeting BPD. Their call for symptom-based targeting and personalized neuromodulation strategies reflects an advanced understanding of both clinical heterogeneity and emerging precision-medicine approaches in psychiatry.
While the manuscript is overall excellent, there are a few areas where improvements could further strengthen it:
-
Language and Style: Although generally well-written, there are minor stylistic inconsistencies and occasional awkward phrasings (e.g., “the opportunity for a home-based treatment opens this avenue” could be more clearly stated as “this also opens the possibility for home-based treatment”). A final professional language polish would enhance clarity and flow, particularly in the discussion and conclusion sections.
-
Psychotherapeutic Integration: The discussion on combining tDCS with psychotherapy is well-conceived, though the section would benefit from more concrete proposals or theoretical models supporting how specific psychotherapeutic elements (e.g., emotion regulation modules of DBT) might synergize with neuromodulation. The reference to guided imagery and cognitive training is intriguing and deserves further elaboration.
-
Critical Evaluation of Literature: Although the manuscript discusses the lack of high-quality RCTs, it could be strengthened by a clearer breakdown of the existing trials' methodological limitations (e.g., sample size, control conditions, blinding procedures). A supplementary table summarizing these studies and their risk of bias would be a valuable addition.
-
Terminological Precision: In some instances, terminology such as "clinical improvements" is used broadly. Clarifying whether these refer to symptom scores, functional outcomes, or both would increase interpretive clarity.
-
Long-Term Outcomes: The section on sustainability of effects is conceptually sound but would benefit from citations to longitudinal studies in related disorders (e.g., depression) where tDCS effects have been shown to persist or decline over time.
Author Response
Reviewer 1.
COMMENT 1: This narrative review presents a timely and clinically significant exploration of the potential role of transcranial direct current stimulation (tDCS) in the treatment of Borderline Personality Disorder (BPD). The manuscript is grounded in a solid understanding of BPD’s neurobiological underpinnings and therapeutic challenges, offering a compelling rationale for considering tDCS as a low-cost, non-invasive neuromodulatory adjunct or alternative to traditional interventions.
We thank the reviewer for emphasizing the key message of this narrative review.
COMMENT 2: The structure of the manuscript is coherent, with well-defined sections covering the pathophysiology of BPD, the neurophysiological mechanisms of tDCS, clinical safety, cost-effectiveness, and preliminary efficacy data. The authors provide a comprehensive and well-referenced synthesis of existing literature, making judicious use of both clinical and preclinical findings. The abstract is well-written and summarizes key points clearly. Moreover, the manuscript adheres to the SANRA guidelines for narrative reviews, enhancing its methodological transparency.
We appreciate the reviewer’s recognition of the methodological rigor and the well-organized structure of the manuscript.
COMMENT 3: The discussion of tDCS mechanisms is neurobiologically sound and includes current evidence on the modulation of fronto-limbic circuits, stress reactivity, and executive function, all of which are relevant to BPD pathophysiology. The authors demonstrate a nuanced understanding of these systems and emphasize the differential impact of electrode montages (e.g., left vs. right DLPFC, rVLPFC), which is a strength of the paper. The figure illustrating common electrode placements is particularly helpful and enhances the clinical applicability of the review.
We sincerely thank the reviewer for highlighting the relevance and clarity of our discussion on tDCS mechanisms, electrode montages, and the clinical utility of the illustrative figure.
COMMENT 4: The emphasis on the safety profile of tDCS is highly appropriate, especially considering the elevated risk of self-harm and suicidality in the BPD population. The comparison to pharmacological alternatives is balanced and data-driven, acknowledging both the limitations of current pharmacotherapy and the absence of FDA-approved medications for BPD. Similarly, the cost analysis is clearly articulated and underscores the potential economic benefits of tDCS, particularly in home-based applications.
We thank the reviewer for emphasizing the relevance of tDCS’s safety profile, which we agree is a significant advantage, especially in the context of BPD. Both clinicians and researchers should carefully consider us.
COMMENT 5: The manuscript’s analysis of tDCS efficacy for core BPD symptoms—emotional dysregulation, impulsivity, and rejection sensitivity—is both critical and well-supported. The authors appropriately qualify their conclusions given the scarcity of randomized controlled trials (RCTs) specifically targeting BPD. Their call for symptom-based targeting and personalized neuromodulation strategies reflects an advanced understanding of both clinical heterogeneity and emerging precision-medicine approaches in psychiatry.
We thank the reviewer for acknowledging our critical analysis and support for personalized, symptom-based neuromodulation approaches in BPD.
COMMENT 6: While the manuscript is overall excellent, there are a few areas where improvements could further strengthen it: Language and Style: Although generally well-written, there are minor stylistic inconsistencies and occasional awkward phrasings (e.g., “the opportunity for a home-based treatment opens this avenue” could be more clearly stated as “this also opens the possibility for home-based treatment”). A final professional language polish would enhance clarity and flow, particularly in the discussion and conclusion sections.
We have addressed the reviewer’s comment on the writing by editing the entire manuscript.
COMMENT 7: Psychotherapeutic Integration: The discussion on combining tDCS with psychotherapy is well-conceived, though the section would benefit from more concrete proposals or theoretical models supporting how specific psychotherapeutic elements (e.g., emotion regulation modules of DBT) might synergize with neuromodulation. The reference to guided imagery and cognitive training is intriguing and deserves further elaboration.
We thank the reviewer for highlighting this critical point, which we agree is essential for both clinical application and future research. Unlike rTMS, tDCS modulates brain activity in a state-dependent manner, meaning it influences the neural circuits that are active during stimulation. In the context of BPD, this opens promising avenues for targeting specific functions such as emotion regulation or executive control during concurrent psychotherapeutic tasks. Following the reviewer’s suggestion, we have expanded this section to include more concrete proposals and theoretical models, particularly regarding the integration of emotion regulation modules from DBT and cognitive training paradigms.
Page 8: “To investigate potential synergies between tDCS and psychotherapy, we propose integrating a symptom-targeted approach. For example, activating emotionally regulated scenarios during anodal stimulation of the left DLPFC may help enhance the neural circuitry underlying emotion regulation. In this context, practicing specific DBT emotion regulation skills—such as emotion labeling—during stimulation could strengthen this domain. Similarly, skills targeting impulsivity, such as the well-known DBT “STOP” technique, could be combined with anodal stimulation of the right DLPFC to reduce impulsive behaviors. Moreover, considering the neuropsychological deficits associated with BPD, it may be beneficial to incorporate cognitive training modules focused on executive functions or working memory (64). Finally, very brief psychotherapeutic interventions, such as guided imagery or imagery rescripting (40), could also be well-suited to the short duration of neuromodulation sessions and offer promising avenues for synergistic treatment.”
COMMENT 8: Critical Evaluation of Literature: Although the manuscript discusses the lack of high-quality RCTs, it could be strengthened by a clearer breakdown of the existing trials' methodological limitations (e.g., sample size, control conditions, blinding procedures). A supplementary table summarizing these studies and their risk of bias would be a valuable addition.
COMMENT 9: Terminological Precision: In some instances, terminology such as "clinical improvements" is used broadly. Clarifying whether these refer to symptom scores, functional outcomes, or both would increase interpretive clarity.
We agree that a more straightforward overview of the existing literature was warranted. Following the reviewer’s recommendation, we have included a table to enhance clarity and synthesis.
|
Author |
Participants |
Anode |
Cathode |
Anodal eectrode size (cm2) |
Current intensity (mA) |
Number of session |
Session duration (mn) |
Main effects |
|
Schulze et al., 2019 |
48 BPD: 25 sham; 23 active |
Right DLPFC |
Left deltoid |
35 |
1 |
1 |
20 |
No improvement of cognitive control over negative stimuli |
|
Molavi et al., 2020 (42) |
32 BPD: 16 sham; 16 active |
Right DLPFC |
Left DLPFC |
25 |
2 |
10 |
20 |
Improvement of executive functions and cognitive emotion regulation, although emotional expression was unaffected |
|
Lisoni et al., 2020 (44) |
30 BPD: 15 sham; 15 active |
Left DLPFC |
Right DLPFC |
35 |
2 |
15 |
20 |
Improvement of impulsivity, aggression, and craving, while also marginally improving decision-making. |
|
Wolkenstein et al. 2020 (43) |
40: 20 healthy; 20 BPD |
Left DLPFC |
Right Mastoïd |
35 |
1 |
1 |
20 |
No reduction of emotional interference at the group level, but selectively improved cognitive control in participants—particularly BPD patients. |
|
Lisco et al., 2025 (47) |
40 BPD: 20 sham; 20 active |
Right VLPFC |
Left supraorbital region |
25 |
1,5 |
|
20 |
Reduction of rejection-related emotions |
COMMENT 10: Long-Term Outcomes: The section on sustainability of effects is conceptually sound but would benefit from citations to longitudinal studies in related disorders (e.g., depression) where tDCS effects have been shown to persist or decline over time.
We agree with the reviewer. While no longitudinal studies have yet examined the durability of tDCS effects in BPD, we have revised the paragraph to include references to studies in depression, where longer-term effects have been observed and appear to be influenced by the number of sessions. These findings support the need for future longitudinal research in BPD to assess the sustainability of clinical benefits and potential relapse.
Page 9: “A critical unanswered question is whether the clinical benefits of tDCS persist beyond the treatment period. Based on recent studies in patients with depression (38), we may hypothesize that longer-lasting effects are more likely to occur with an increased number of sessions. Longitudinal studies are needed to evaluate the durability of symptom improvement and the risk of relapse over time.”
Reviewer 2 Report
Comments and Suggestions for Authors
Thanks for the invitation to review this interesting manuscvript.
The abstract is well structured and gives a good summary of the topic. However, I would like to suggest that instead of saying “comprehensive literature synthesis,” the authors could briefly write what the search criteria or systematic inclusion criterias were for selecting the literature. I also advise the authors to say something about the number of included studies or the design of the main trials they found.
The introduction is good and explains clearly why tDCS could be a relevant and innovative treatment method. The rationale is well described. Still, there is no explicit research question or hypotheses. I think the authors should add this. Even in a narrative review, it's helpful to include some explicit hypothesis (like “We assume that...”) to guide how the literature is interpreted. The literature used is mostly up to date. What I missed though was a clear inclusion or exclusion strategy: there is no PRISMA diagram or any overview like that — unless I overlooked it. This should be added, because otherwise it's unclear how studies were selected. Also, some of the claims (like about cost-effectiveness) are based on economic data that might not be relevant for health care systems outside of France or Canada. More general or context-diverse data would make the argument stronger.
The methods section is a weak point. It says the review followed the SANRA guidelines, which is a good thing. But the description of the literature search is quite brief. There is no mention of inclusion criteria, how the authors judged the studies, or how they selected the data. Also, there’s no form of bias control — unless the authors think this is not needed in a narrative review, but then they should explain why.
There is no clear summary of individual studies, with details like sample size, setting, duration, design, and outcomes. There is also no table showing all the included studies. Some studies are mentioned (like Molavi et al. or Lisco et al.), but it stays unclear how many studies were actually reviewed and how strong the conclusions are.
The part about tDCS itself is well done. The authors describe the intensity, duration, electrode placement, and the difference between anodal and cathodal stimulation. This part is clear and informative. It’s also nice that the montage options are shown with a figure.
The results are organized in clear sub-sections, which makes it easier to read and understand the effects for different symptoms. But the effect sizes are almost never given, and statistical results (like p-values or CI’s) are not mentioned — unless, again, this is normal for a narrative review, but then that should be said clearly. Otherwise, the effects (e.g. for impulsivity) remain vague and mostly qualitative.
The discussion reflects well on the existing literature and doesn't just repeat the results. It’s good that the authors talk about the limitations of tDCS and that they clearly mention the lack of well-designed RCTs. But they don’t discuss alternative explanations when no effect was found in some studies — for example, placebo effects or wrong target groups.
There is a section about limitations of the current evidence (lines 304–319), which is positive. Still, the overall tone of the conclusion is quite optimistic, even though the evidence is weak, as the authors themselves noted. A stricter, more careful final conclusion would be more appropriate.
Author Response
Reviewer 2
Thanks for the invitation to review this interesting manuscvript.
COMMENT 1: The abstract is well structured and gives a good summary of the topic. However, I would like to suggest that instead of saying “comprehensive literature synthesis,” the authors could briefly write what the search criteria or systematic inclusion criterias were for selecting the literature. I also advise the authors to say something about the number of included studies or the design of the main trials they found.
We acknowledge that the methodology in the abstract needed to be more explicit. In response to the reviewer’s suggestion, we have fully revised the Methods section to clarify our search strategy, inclusion approach, and the number and type of studies included.
Page 1: “We conducted a narrative literature synthesis in accordance with the SANRA (Scale for the Assessment of Narrative Review Articles) guidelines. A PubMed/MEDLINE search was performed using keywords related to transcranial direct current stimulation (tDCS) and BPD, identifying five published randomized controlled trials on the topic. To provide a broader perspective, we also included studies from related fields examining mechanisms of action, safety and tolerability, cost-effectiveness, stimulation parameters, and clinical outcomes relevant to BPD.”
COMMENT 2: The introduction is good and explains clearly why tDCS could be a relevant and innovative treatment method. The rationale is well described. Still, there is no explicit research question or hypotheses. I think the authors should add this. Even in a narrative review, it's helpful to include some explicit hypothesis (like “We assume that...”) to guide how the literature is interpreted.
In line with the reviewer’s suggestion, we have thoroughly revised the final section of our Introduction to articulate our research question and hypotheses clearly.
Page 3: “In this context, this narrative review aims to explore the potential role of tDCS in the treatment of BPD. Specifically, it will examine the rationale for integrating tDCS into BPD care—focusing on its safety, affordability, and capacity to target key symptom domains such as emotional dysregulation and impulsivity—and will discuss optimal stimulation protocols (electrode placement, intensity, session duration, and frequency) as well as its potential to complement or enhance psychotherapy. We ask whether tDCS can safely, affordably, and effectively modulate core BPD features, and we hypothesize that: (1) specific tDCS parameters (e.g., electrode montage and stimulation intensity) will differentially influence symptomatic targets, from core BPD symptoms to executive functions and common comorbidities; and (2) tDCS will provide additional benefits in terms of cost-effectiveness and synergistic gains when combined with psychotherapeutic interventions. Finally, we will address clinical applications and feasibility, evaluate tDCS as both a standalone and adjunctive treatment, highlight current gaps in the literature, and outline future research directions.”
.
COMMENT 3: The literature used is mostly up to date. What I missed though was a clear inclusion or exclusion strategy: there is no PRISMA diagram or any overview like that — unless I overlooked it. This should be added, because otherwise it's unclear how studies were selected.
The reviewer is correct: we did not include a PRISMA flow diagram, and our description of the study-selection process was not detailed enough to be reproducible. In fact, our review proceeded in two phases:
- Phase 1 – Systematic Review of tDCS in BPD: We conducted a comprehensive search for clinical trials involving tDCS in patients with BPD. Midway through this work, another systematic review on the same topic was published.
- Phase 2 – Targeted “secondary” searches: Building on that prior review, we conducted focused searches on three key areas: Economic evaluations of tDCS (cost studies), Safety assessments (clinical guidelines and meta-analyses), Impact on common BPD comorbidities (dedicated meta-analyses)
While this two-step, narrative-style approach is acceptable under SANRA’s moderate-quality criteria, we agree that transparency and reproducibility demand more. We have therefore revised the Methods section.
Page 3: “The review process was carried out in two steps. First, we focused on identifying RCTs investigating tDCS in individuals with BPD. Data extraction was conducted independently by two reviewers (MB, LC), based on a prior unpublished systematic review conducted by our team. Second, we performed targeted searches to complement this initial synthesis by including studies addressing: (1) the efficacy of tDCS in treating common BPD comorbidities (identified through recent meta-analyses), (2) the safety profile of tDCS (via meta-analyses and clinical guidelines), and (3) cost-related aspects (medico-economic evaluations). Finally, we incorporated selected studies to enrich our understanding of the neurobiological mechanisms underlying tDCS and its potential synergies with psychotherapeutic interventions. »
COMMENT 4: Also, some of the claims (like about cost-effectiveness) are based on economic data that might not be relevant for health care systems outside of France or Canada. More general or context-diverse data would make the argument stronger.
The reviewer’s comment is entirely valid—we acknowledge that the data we present are limited in scope, particularly regarding the broader economic implications of tDCS. This is a significant limitation, as tDCS is a relatively low-cost intervention compared to long-term psychotherapy, pharmacological treatments, or even rTMS. Its affordability may offer substantial benefits in low-resource settings or countries facing economic constraints. We have re-examined the available literature to identify any additional data on this topic but found none. Therefore, we propose to explicitly highlight this gap in the literature as a limitation in the revised manuscript.
Page 5:” It is important to note that existing economic evaluations of tDCS have been conducted in only a few countries—primarily France and Canada—and therefore cannot be readily generalized to an international context. There is a clear lack of global data. Moreover, tDCS devices typically require approval from national regulatory authorities, which can lead to substantial variability in availability, cost, and implementation across countries. The current literature does not address these discrepancies. Device pricing may differ significantly depending on the region, and the overall cost-effectiveness of tDCS is closely tied to the structure and resources of the local healthcare system. As such, caution is warranted when interpreting or extrapolating cost-effectiveness data for broader use.”
COMMENT 5: The methods section is a weak point. It says the review followed the SANRA guidelines, which is a good thing. But the description of the literature search is quite brief. There is no mention of inclusion criteria, how the authors judged the studies, or how they selected the data. Also, there’s no form of bias control — unless the authors think this is not needed in a narrative review, but then they should explain why.
We thank the reviewer for this valuable observation. While narrative reviews are not held to the same procedural standards as systematic reviews, we fully agree that enhancing transparency strengthens the manuscript’s overall quality and credibility. Accordingly, we adhered to the SANRA guidelines for narrative reviews instead of the PRISMA guidelines, which are specific to systematic reviews.
In response, we have expanded the Methods section to clarify our approach (see response to Comment 3). However, we did not include a PRISMA flow diagram, as our process did not involve a systematic selection with predefined exclusion criteria—beyond general inclusion parameters such as publication date. Consistent with the SANRA guidelines, we also did not conduct a formal quality appraisal of the included studies. If the reviewer deems it necessary, we could explicitly add this information in the manuscript: namely, that no formal exclusion criteria were applied, no PRISMA diagram was used given the narrative nature of the review, and no quality assessment was conducted due to the absence of pooled calculations and the broader, exploratory scope of the review.
COMMENT 6: There is no clear summary of individual studies, with details like sample size, setting, duration, design, and outcomes. There is also no table showing all the included studies. Some studies are mentioned (like Molavi et al. or Lisco et al.), but it stays unclear how many studies were actually reviewed and how strong the conclusions are.
We agree that a table would improve the reader’s overview of the literature by providing a clear and systematic summary of the included studies. We have now added this table to the manuscript.
COMMENT 7: The part about tDCS itself is well done. The authors describe the intensity, duration, electrode placement, and the difference between anodal and cathodal stimulation. This part is clear and informative. It’s also nice that the montage options are shown with a figure.
We thank the reviewer for this positive comment. We're pleased that the tDCS section and the illustrative figure were found clear and informative.
COMMENT 8: The results are organized in clear sub-sections, which makes it easier to read and understand the effects for different symptoms. But the effect sizes are almost never given, and statistical results (like p-values or CI’s) are not mentioned — unless, again, this is normal for a narrative review, but then that should be said clearly. Otherwise, the effects (e.g. for impulsivity) remain vague and mostly qualitative.
The reviewer rightly highlights the lack of quantitative data in the Results section. This is indeed a challenge, as we did not conduct a systematic review and therefore did not assess study quality or perform pooled effect size calculations. However, we have now added relevant quantitative information, such as effect sizes or statistical outcomes, when available from key sources, to illustrate better and support the reported effects.
Page 6: “As an illustration, we calculated the effect size for emotion regulation (measured by the Emotion Regulation Questionnaire) in the Molavi study, which yielded a very large effect (Cohen’s d = 4).
As an illustration, we calculated the effect size for impulsivity, measured by the Barratt Impulsiveness Scale (BIS-11), in the Lisoni study, which showed a large effect (Cohen’s d = 1.12).
As an illustration, we calculated the effect sizes for rejection-related emotions, measured by the Rejected Emotion Scale (RES), in the Lisco study. The results showed a large effect for inclusion (Cohen’s d = 0.95), a moderate effect for exclusion (Cohen’s d = 0.75), and no significant effect for overinclusion. “
COMMENT 9: The discussion reflects well on the existing literature and doesn't just repeat the results. It’s good that the authors talk about the limitations of tDCS and that they clearly mention the lack of well-designed RCTs. But they don’t discuss alternative explanations when no effect was found in some studies — for example, placebo effects or wrong target groups.
We fully agree with the reviewer. As emphasized in Lisoni's systematic review, advancing the field requires studies with better design and adequate power. Without this, there is a significant risk of generating false positives or overlooking meaningful clinical improvements, both of which could hinder a clear understanding of the true potential of this technique in the treatment of BPD.
Page 6: “Several studies failed to demonstrate an effect on their primary outcomes. This may be explained by various factors, including methodological limitations (e.g., underpowered designs), population heterogeneity (e.g., medication use, comorbidities, severity levels), suboptimal stimulation parameters (e.g., dosage, electrode placement, number of sessions), or insufficient control of brain activity during stimulation. »
COMMENT 10: There is a section about limitations of the current evidence (lines 304–319), which is positive. Still, the overall tone of the conclusion is quite optimistic, even though the evidence is weak, as the authors themselves noted. A stricter, more careful final conclusion would be more appropriate.
We agree with the reviewer and acknowledge the importance of distinguishing between core BPD symptoms and comorbid conditions, particularly depression. In response, we have added clarifying information to highlight this distinction and better reflect the varying levels of evidence supporting the use of tDCS in these domains.
Page 10: “We could state that while several comorbidities, such as depression, are supported by strong evidence that may justify clinical implementation of tDCS, the effects on core BPD features such as impulsivity, affective instability, and rejection sensitivity remain preliminary and require confirmation through more robust, well-powered studies.”